# Antioxidant and Hepatoprotective Potential of *Echinops ritro* L. Extracts on Induced Oxidative Stress In Vitro/In Vivo

**DOI:** 10.3390/ijms24129999

**Published:** 2023-06-10

**Authors:** Dimitrina Zheleva-Dimitrova, Rumyana Simeonova, Magdalena Kondeva-Burdina, Yonko Savov, Vessela Balabanova, Gokhan Zengin, Alexandra Petrova, Reneta Gevrenova

**Affiliations:** 1Department of Pharmacognosy, Faculty of Pharmacy, Medical University-Sofia, 1000 Sofia, Bulgaria; vbalabanova@pharmfac.mu-sofia.bg (V.B.); rgevrenova@pharmfac.mu-sofia.bg (R.G.); 2Department of Pharmacology, Pharmacotherapy, and Toxicology, Faculty of Pharmacy, Medical University-Sofia, 1000 Sofia, Bulgaria; mkondeva@pharmfac.mu-sofia.bg (M.K.-B.); alexpetrova.work@gmail.com (A.P.); 3Institute of Emergency Medicine “N. I Pirogov”, Bul. Totleben 21, 1606 Sofia, Bulgaria; yonko_savov@hotmail.com; 4Physiology and Biochemistry Research Laboratory, Department of Biology, Science Faculty, Selcuk University, Campus, Konya 42130, Turkey; gokhanzengin@selcuk.edu.tr

**Keywords:** *Echinops ritro*, hepatoprotective potential, oxidative stress, secondary metabolites, UHPLC-HRMS

## Abstract

*Echinops ritro* L. (Asteraceae) is traditionally used in the treatment of bacterial/fungal infections and respiratory and heart ailments. The aim of this study was to evaluate the potential of extracts from *E. ritro* leaves (ERLE) and flowering heads (ERFE) as antioxidant and hepatoprotective agents on diclofenac-induced lipid peroxidation and oxidative stress under in vitro and in vivo conditions. In isolated rat microsomes and hepatocytes, the extracts significantly alleviated oxidative stress by increasing cell viability and GSH levels and reducing LDH efflux and MDA production. During in vivo experiments, the administration of the ERFE alone or in combination with diclofenac resulted in a significant increase in cellular antioxidant protection and a decrease in lipid peroxidation witnessed by key markers and enzymes. A beneficial influence on the activity of the drug-metabolizing enzymes ethylmorphine-N-demetylase and aniline hydroxylase in liver tissue was found. In the acute toxicity test evaluation, the ERFE showed no toxicity. In the ultrahigh-performance liquid chromatography–high-resolution mass spectrometry analysis, 95 secondary metabolites were reported for the first time, including acylquinic acids, flavonoids, and coumarins. Protocatechuic acid *O*-hexoside, quinic, chlorogenic and 3, 5-dicaffeoylquinic acid, apigenin; apigenin 7-*O*-glucoside, hyperoside, jaceosidene, and cirsiliol dominated the profiles. The results suggest that both extracts should be designed for functional applications with antioxidant and hepatoprotective capacity.

## 1. Introduction

The liver, as the most important organ in the biotransformation and detoxification of remedies, produces oxygen free radicals, which are necessary byproducts for an aerobic existence [1]. A high number of prescription and over-the-counter drugs, herbal products, and food additives can cause liver toxicity due to oxidative stress [2]. Reactive oxygen species (ROS) have long been implicated in the pathophysiology of liver injury. However, in pathophysiological conditions, either due to the liver metabolism of remedies, the obstruction of the bile duct or conditions of ischemia, the balance between the generation of free radicals and the capacity of the liver to detoxify them can be shifted such that oxidative stress can negatively influence hepatic homeostasis and hepatocyte function [2].

Diclofenac (DF) is a widely used medication from the group of non-steroidal anti-inflammatory drugs (NSAIDs). Nevertheless, its usage has been associated with the occurrence of some adverse drug reactions, including liver damage. Several studies [3,4] have suggested that the oxidative bioactivation of DF by cytochrome P450 isoenzymes (CYPs) plays a role in liver toxicity. These bioactivation reactions lead to the formation of several reactive metabolites, including *p*-benzoquinone imines, which, in addition, could be inactivated by conjugation with reduced glutathione (GSH).

Numerous plant extracts and their specialized metabolites have demonstrated the potential to ameliorate DF-induced hepatotoxicity via several mechanisms, the most debated and studied mechanism being the antioxidant potential of natural products [5,6,7,8].

The genus *Echinops* L. (Asteraceae family, Cardueae tribe) comprises more than 120 species, which are distributed worldwide [9,10,11]. The main distinguished feature of the taxon is the presence of uniflowered capitula aggregated into second-order spherical or oval heads [12].

Based on the literature survey on chemical profiles, *Echinops* roots are mostly characterized by the occurrence of thiophenes such as α-terthiophene, grijisyne A, echinopsacetylenes A and B, etc. In one study, flavonoids were isolated from the root of *E. grijsii* and the whole plant of *E. echinatus*. It was shown that these flavonoids might be responsible for the hepatoprotective effects of the extracts [13]. Methotrexate-induced hepatotoxicity was also used to evaluate the hepatoprotective effect of *E. heterophyllus* extract from aerial parts and flavonoid fraction in rabbits. The ethanolic extract at a dose of 250 mg/kg significantly decreased the serum proteins, liver enzymes, and oxidative stress markers compared to the flavonoid fraction [14]. The methanol root extract of *E. giganteus* showed in vitro a free radical scavenging effect with 12.54 mg Trolox equivalent per 100 g [15]. The aqueous and ethanol extracts of *E. ritro* and *E. tournefortii*, as well as the methanol-ethyl acetate extract of *E. persicus,* showed a significant DPPH free radical scavenging effect [16,17,18,19]. Regardless of the effects described, the antioxidant activity evaluations are still not sufficient. No single in vivo antioxidant model was employed. 

Nevertheless, the aerial parts are rich in terpenes (santamarin, β-amyrin, betulinic acid, taraxasterol, etc.), flavonoids and other phenolic compounds (apigenin, luteolin, nivegin, 7-hydroxyisoflavone, kaempferol, etc.). Additionally, essential oil and alkaloids are presented in all plant parts [11].

In accordance with the chemical composition, *Echinops* species have been used traditionally in the treatment of different diseases including bacterial/fungal infections, fever, respiratory and heart ailments, etc. [11]. The phytochemical characteristics of *Echinops* sp. and an in vitro screening revealed antioxidant, anti-microbial, and therapeutic activities for an immunomodulatory purpose [20].

The presence of thiophenes defines antifungal and antibacterial, cytotoxic, anti-malarial and insecticidal pharmacological activity. Terpenes as well as flavonoids and other phenolic compounds are responsible for anti-inflammatory, antioxidant and hepato-protective effects [11,21]. Most of the studies were performed on tetrachlormethane (CCl_4_)-induced liver damage, in which the biomarkers of liver function, ASAT and ALAT, were measured [11]. Aqueous and butanol extracts of *E. grijsii* root, at a dose of 300 mg/kg, and ethanol extracts of the aerial parts of *E. echinatus* at 500 and 750 mg/kg resulted in a significant decrease in ALAT and ASAT. 

Herein, *Echinops ritro* L. (southern globethistle) methanol aqueous extracts were investigated. The species is a herbaceous perennial plant widely distributed across dry and stony habitats up to 1000 m a.s.l. and is native to Southern and Eastern Europe and Western Asia [22,23]. A recent work highlighted a high percentage of mono- and di-caffeoylquinic acids in the *E. ritro* aerial parts (leaves and seeds) without differentiation between respective isomers and a scarce number of flavonoids [24]. Although *Echinops* species are drawing increased interest from phytochemists, much of the research focus has been centered on the thiophenes and terpenes [11]. It is worth noting that there is no in-depth study on the *E. ritro* polyphenolic compounds by hyphenated platform ultra-high-performance liquid chromatography–high resolution mass spectrometry (UHPLC-HRMS) integrated with an assessment of in vitro and in vivo protective effects in induced lipid peroxidation and oxidative stress injury. Despite the accumulating evidence that *Echinops* extracts reduce liver enzymes and antioxidant markers [11], the effects of any *Echinops* species on rat liver microsomes and hepatocytes and the histological profile of in diclofenac-induced liver oxidative stress have not been thoroughly investigated. Notwithstanding studies on the chemical composition of *E. ritro*, an additional detailed investigation of both secondary metabolites and biological potential appears necessary. The antioxidant and hepatoprotective potential of the extracts derived from *E. ritro* leaves (ERLE) and flowering heads (ERFE) combined with phytochemical profiling reveals a new insight into the taxa for its application for health benefits.

## 2. Results

### 2.1. Spectrophotometric Determination of Antioxidant Activity of ERFE and ERLE 

Based on the various spectrophotometric tests performed to define the antioxidant profile of both extracts, DPPH, ABTS, CUPRAC and FRAP activities were reported exclusively higher for *E. ritro* leaf extract (ERLE) (120.13 ± 0.38 mgTE/g, 136.48 ± 1.26 mgTE/g, 251.67 ± 9.28 mgTE/g and 153.54 ± 8.27 mgTE/g, respectively) compared to those of the flowering head extract (ERFE) (Table 1). These findings are in accordance with the results for the total phenolic and flavonoid contents. Nevertheless, the data for metal chelating and phosphomolybdenum inhibitory potential revealed a similar activity for both extracts, as follows: leaves: 29.22 ± 0.22 mg EDTAE/g and 1.52 ± 0.03 mmol TE/g, respectively; flowering heads: 26.51 ± 0.55 mg EDTAE/g and 1.24 ± 0.01 mmol TE/g, respectively (Table 1).

### 2.2. In Vitro Studies on Antioxidant Activity of ERFE and ERLE in Rat Liver Microsomes and Hepatocytes

The incubation of rat liver microsomes and hepatocytes with the ERFE and ERLE at two concentrations (10 µg/mL and 50 µg/mL) did not demonstrate statistically significant hepatotoxic and pro-oxidant effects (Figure 1 and Figure 2). Under the conditions of non-enzyme-induced lipid peroxidation (iron sulphate/ascorbic acid, Fe/AA), on liver microsomes, both extracts showed statistically significant, concentration-dependent antioxidant effects similar to the effects of the classical antioxidant silymarin (Figure 1). Fe/AA significantly increased malon dialdehyde (MDA) production by 150% compared to the control (untreated microsomes) (Figure 1). Both low concentrations of the ERFE and ERLE (LC, 10 µg/mL) and high (HC, 50 µg/mL) concentrations of the ERFE and ERLE significantly reduced MDA production by 78%, 84%, 72% and 74%, respectively, compared to the toxic agent Fe/AA (Figure 1). These results are similar to the effect of silymarin, which reduced MDA production by 80% and 82% for low (10 µg/mL) and high (50 µg/mL) concentrations, respectively.

In the *tert*-butyl hydroperoxide (t-BuOOH)-induced oxidative stress model in rat hepatocytes, the ERFE and ERLE showed good, statistically significant and concentration-dependent hepatoprotective effects compared with those of *tert*-butyl hydroperoxide (Figure 3 and Figure 4). The effects of the ERFE were more pronounced than those of the ERLE and similar to those of silymarin.

The incubation of hepatocytes with t-BuOOH significantly reduced cell viability by 60% and the GSH level by 50% compared to the control (untreated rat hepatocytes) (Figure 3). The lower ERFE concentration increased cell viability by 50% and the GSH level by 40% compared to the toxic agent (t-BuOOH) (Figure 3). The higher concentration of the ERFE greatly increased cell viability by 75% and the GSH level by 60% compared to t-BuOOH. The low dose of the ERLE significantly increased cell viability by 25% and the GSH level by 20% compared with t-BuOOH.

The high dose (50 µg/mL) of the ERLE also increased cell viability by 50% and the GSH level by 30% compared to the toxic agent. Both concentrations (LC and HC) of silymarin notably increased cell viability by 50% and 75%, respectively, and the GSH level by 40% and 60% compared with the toxic agent (t-BuOOH) (Figure 3).

The incubation of hepatocytes with t-BuOOH considerably increased lactate dehydrogenase (LDH) leakage by 250% and MDA production by 150% compared with the control (untreated rat hepatocytes). Both the LC and HC ERFE concentrations reduced LDH leakage by 43% and 71% and MDA production by 60% and 80%, respectively, compared to the toxic agent. Both the LC and HC of the ERLE also significantly reduced LDH leakage by 29% and 57% and MDA production by 20% and 40%, respectively, compared to t-BuOOH. Silymarin produced the same effects. Both concentrations significantly reduced LDH leakage by 43 and 71% and MDA production by 60 and 80%, respectively, compared to the toxic agent t-BuOOH (Figure 4).

### 2.3. In Vivo Assessment of the Effects of ERFE

The antioxidant effects of the ERFE were more pronounced in in vitro experiments than the effects of the ERLE. For this reason, all in vivo experiments were performed with the ERFE.

#### 2.3.1. Acute Toxicity Test in Rats

In the acute toxicity test, doses of 3000, 2000 and 1000 mg/kg of ERFE were administered orally to male rats. All three doses produced no side or toxic effects in the experimental animals during the first day of oral administration or until the end of the observation period (fourteenth day). Based on these results, the LD_50_ value after oral administration is estimated to be higher than 3000 mg/kg. A dose of 300 mg/kg (≈1/10 of the LD_50_) was chosen for the repeated administration of the extract in the evaluation of its hepatoprotective effects under in vivo conditions. The histological examination of liver sections from the three dose groups of rats showed no major pathological changes in hepatocyte morphology (Figure 5).

#### 2.3.2. Effects of ERFE on Diclofenac-Induced Liver Oxidative Stress in Rats

##### Assessment of Biochemical Markers in Rat Serum

The results from serum biochemical evaluation are presented in Table 2. Fourteen days of the oral administration of the ERFE alone did not influence the investigated parameters compared to the control group and reference values. The oral administration of 50 mg/kg of DF for three days slightly increased the blood concentrations of urea, total and direct bilirubin, triglycerides and the activity of ASAT and GGT compared to the controls, but these parameters remained close or similar to the reference values. In the combination group (ERFE + DF), increased GGT activity and triglyceride concentration were seen compared to the controls but were within the reference range. 

##### Lipid Peroxidation, Reduced Glutathione and Antioxidant Enzymes Activity

The effect of the ERFE on lipid peroxidation and antioxidant profile in rats with diclofenac-induced oxidative stress is shown in Table 3.

The repeated oral administration of the ERFE alone did not change the MDA level but did increase the GSH level by 15% compared to the control value. With respect to control animals, diclofenac resulted in an increased amount of MDA by 26% (*p* < 0.05) and a decreased GSH level by 22% (*p* < 0.05). In the combined group, fourteen days of administration of the extract and three days of oral diclofenac treatment resulted in a significant decrease in the amount of MDA by 18% (*p* < 0.05) and an increase in GSH levels by 23% compared to the group treated with the hepatotoxic agent alone. 

Oral treatment with the ERFE alone induced a considerable antioxidant effect in terms of increasing GPx, GR and GST activity by 15, 17 and 12%, respectively, compared to the controls. Three days of diclofenac (50 mg/kg) alone resulted in a significant decrease (*p* < 0.05) in GPx activity by 10%, GR by 21% and GST by 17% compared to the control animals. In the combination group, there was a prominent (*p* < 0.05) increase in GPx activity by 16%, GR by 27% and GST by 21% compared to the group treated with diclofenac alone.

##### Effects on Ethylmorphine-N-demetylase (EMND) and Aniline Hydroxylase (AH) Activity

Table 4 shows the effects of the tested compounds, administered alone and in combination, on EMND and AH activity. Administered alone, diclofenac increased the activity of both enzymes by 14% and 12%, respectively, compared to the controls. The ERFE alone showed some inhibitory effects, as evidenced by the reductions in EMND and AH activity by 22% (*p* < 0.05) and 14% (*p* < 0.05), respectively. This inhibitory effect of the extract was also recorded in the combination group, in which the studied enzyme activity decreased significantly by 25% and 18%, respectively, compared to the DF only group.

### 2.4. UHPLC-HRMS Profiling of Specialized Metabolites in E. ritro Extracts

Based on the retention times, the MS and MS/MS accurate masses, the fragmentation patterns in the MS/MS spectra, the relative ion abundance and a comparison with reference standards and the literature, a total of 108 specialized metabolites were identified or tentatively annotated in the *E. ritro* extracts (Table 5). The compounds were identified as level 1 (metabolites that compare to authentic standards) and level 2 (all others). The total ion chromatograms of both extracts in a negative ion mode are presented in Appendix A.

Hydroxybenzoic acids, hydroxycinnamic acids, phenylethanoid glycosides, coumarins and acylquinic acids

Hydroxybenzoic acids (3, 8, 9, 10, 13, 14, 15, 18, 27 and 29) and their glycosides (1, 2, 5, 6, 7, 9, 10, 14, 15, 18, 25, 31 and 32), quinic acid (17), shikimic acid (21), and hydroxycinnamic acids (20, 24, 26 and 28) and their glycosides (19 and 30) were identified based on the comparison of retention times, the exact masses and fragment spectra with reference standards and the literature (Appendix A) [26]. The MS/MS spectra of sugar esters syringyl O-hexose (12) and caffeoyl O-hexose (16) were acquired. In contrast to the corresponding hexosides, fragment ions resulting from the sugar cross ring cleavages were registered in the sugar esters as follows: ^0,4^Hex (−60 Da), ^0,3^Hex (−90 Da) and ^0,2^Hex (−120 Da) [27]. Compound 22 ([M-H]^–^ at *m/z* 177.0193) gave fragment ions at *m/z* 149.023 and 133.028, resulting from the sequential loss of CO and CO_2_ groups. Thus, the coumarin structure of aesculetin was proposed for compound 22 [28]. A similar MS/MS spectrum was obtained for 11, but, here, an initial loss of hexose was observed, and the compound was identified as aesculetin O-hexoside (Appendix A) [28].

A variety of acylquinic acids (AQA), including 10 mono-, 13 di- and 1 triacylquinic acid and 4 hexosides, were identified/annotated in the *E. ritro* extracts (Appendix A). Fragmentation patterns and diagnostic ions in the MS/MS spectra of AQA were reported elsewhere [29,30,31]. Compounds 34, 37, 52, 53, 55, 56 and 60 were identified based on the comparison of retention times, exact masses and fragment spectra with reference standards and the literature (Table 5 and Appendix A).

The extracted ion chromatograms of phenolic acids and derivatives showed that the *E. ritro* leaf profile was dominated by protocatechuic acid-*O*-hexoside (1) (10.09%), gentisic acid (27) (2.26%) and protocatechuic acid-(salicyl)-hexoside (32) (3.05%) together with quinic acid (17) (16.24 %). Among acylquinic acid, the predominant compounds in both the leaves and flower heads were neochlorogenic (34) (6.59% in the ERLE and 0.67% in the ERFE), chlorogenic (37) (18.28% in the ERLE and 9.35% in the ERLE), 4-caffeoylquinic (38) (17.52% in the ERLE and 9.29% in the ERFE) and 3, 5-dicaffeoylquinic acid (53) (9.90% in the ERLE and 18.44% in the ERFE) (Appendix A).

Flavonoids

The strategy for flavonoid assignment was consistent with that reported elsewhere [26,27,30,31]. It was based on MS and MS/MS accurate masses and the conformity of the fragmentation “fingerprints” of different classes’ flavonoids in *Tanacetum* sp. and *Achillea* sp. The neutral losses of 162.054, 176.033 and 308.112 Da correspond to *O*-hexose/hexuronic acid/rutinose, respectively. Based on the comparison with the reference standards, compounds 64, 69, 75, 76 and 77 were identified as isoquercitrin, luteolin 7-*O*-glucoside, astragalin, isorhamnetin 3-*O*-glucoside and apigenin 7-*O*-glucoside, respectively. Accordingly, 63, 65, 72 and 73 were ascribed to rutin, luteolin 7-*O*-rutinoside, kaempferol 3-*O*-rutinoside and isorhamnetin 3-*O*-rutinoside, respectively (Appendix A). The acetylhexososides (71, 79, 82, 86 and 89) and p-coumaroylhexosides (95, 96, 100 and 101) were deduced from the concomitant losses of hexosyl and either acetyl (204.064 Da, C_8_H_12_O_6_) or coumaroyl (308.091 Da, C_15_H_16_O_7_) moieties, respectively. A key point in the flavone and flavonol aglycone annotation was a series of neutral losses of CO (−28 Da), CO_2_ (−44 Da), CH_2_O (−30 Da) and H_2_O (−18 Da), which were supported by the retro-Diels–Alder (RDA) cleavages ^0,4^A^−^, ^1,2^A^−^, ^1,3^A^−^, ^1,2^B^−^ and ^1,3^B^−^ [26,31].

The flavonoid glycosides profile of the leaf extract was dominated by isoquercitrin (64) (11.88%) and hyperoside (67) (11.38%) together with luteolin-*O*-hexuronide (68) (11.28), rutin (63) (6.39%), kaempferol 3-*O-*rutinoside (72) (4.50%), astragalin (75) (4.50%) and apigenin 7-*O*-hexuronide (77) (3.92%). In contrast, 77 (5.57%) is the prevailing flavonoid glycoside in the flower head extract accompanied by apigenin *O*-rutinoside (74) (2.63%) and hyperoside (67) (1.12%) (Appendix A). Previously, apigenin was determined in *E. echinatus*, *E. spinosus* and *E. orientalis* [11], while luteolin 7-*O*-glucoside was found in *E. spinosus* [32].

The strategy for 6-methoxyflavonoids annotation was based on the precursor and indicative fragment ions delineated in the studies on *Tanacetum* sp. [26,27,31]. Thus, patuletin (quercetagetin 6-methyl ether) (88) and hispidulin (scutellarein 6-methyl ether) (91) were identified on the basis of the RDA ions resulting from the neutral and radical losses of [^1,3^A-•CH_3_]^−^, [^1,3^A-•CH_3_-CO]^−^, [^1,3^A-CH_4_]^−^, [^1,3^A-CO-CH_2_]^−^ and [^1,3^A-CO-CH_4_]^−^ (Appendix A). Additionally, the aforementioned compounds were unambiguously identified by comparison with the reference standards. The (−) ESI-MS/MS spectra of two isobars (92 and 99) at *m*/*z* 329.067 [M-H]^−^ were acquired (Appendix A). Cirsiliol (6-hydroxyluteolin-6, 7-dimethyl ether) (92) was deduced from the subsequent losses of methyl radicals at *m/z* 314.044 [M-H-•CH_3_]^−^ and 299.020 [M-H-2•CH_3_]^−^, which were supported by *m/z* 161.023 [^1,3^A-CH_4_-H_2_O]^−^ and 151.003 [^1,3^A-CH_4_-CO]^−^. On the other hand, jaceosidin (6-hydroxyluteolin-6,3′-dimethyl ether) (99) yielded prominent fragment ions indicating 6-methoxylation in the A-ring at *m/z* 165.989 [^1,3^A-CH_3_]^−^, 164.982 [^1,3^A-CH_4_]^−^, 163.003 [^1,3^A-H_2_O]^−^ and 136.986 [^1,3^A-CO-CH_4_]^−^, while the methoxylation in the B-ring was discernable from the fragment at *m/z* 133.028 [^1,3^B-CH_2_]^−^ [31]. The MS/MS spectrum of 105 ([M-H]^−^ at 343.082) was consistent with that of 99, except for the appearance of an additional methyl group in the B-ring deduced from the RDA ions at *m/z* 147.043 [^1,3^B-CH_2_]^−^ and 132.021 [^1,3^B-•CH_3_-CH_2_]^–^, as was observed in eupatilin (6-hydroxyluteolin-6, 3′, 4′-trimethyl ether) or santin (6-hydroxykaempferol-6, 3, 4′-trimethyl ether).

In the same manner, closely related quercetagetin derivatives (94, 97 and 102) were described. Both compounds 94 and 102 showed the same [M-H]^−^ at *m/z* 345.062 indicating an additional methyl group in comparison with 88 (Appendix A). RDA ions generated from [^1,3^A]^−^ were consistent with those in 88. A fragment ion at *m/z* 121.028 [^1,2^B]^–^ in 102 suggested a lack of methoxylatation in the B-ring, as was observed in axillarin (quercetagetin-3, 6-dimethyl ether), while the ion at *m/z* 163.039 [^1,3^B-CH_2_]^−^ suggested the presence of methoxylation in the B-ring, as was seen in quercetagetin-6, 3′-dimethyl ether (spinacetin). Concerning 97, it was ascribed to quercetagetin 3, 6, 3′(4′)-trimethyl ether. Prominent fragment ions at *m/z* 148.015 [^1,3^B-•CH_3_-CH_2_]^−^ and 163.039 (^1,3^B^−^-CH_2_) indicated the presence of two methoxy groups in the RDA ion [^1,3^B]^−^ (Appendix A). Both isomers 106 and 108 at *m/z* 343.083 [M-H]^−^ generated prominent ions at *m/z* 328.059, 313.036 and 299.056, resulting from the consecutive losses of three methyl radicals •CH_3_. These were assigned to quercetin trimethyl ether without methoxylation in C-6, as was observed in ayanin (quercetin 3, 7, 3′-trimethyl ether) (Table 5 and Appendix A). 

Overall, eupatilin (105) (4.60%) together with cirsiliol (92) (3.85%) and jaceosidin (99) (3.71%) appeared to be characteristic of the leaf extract; in contrast, flower heads were the richest in apigenin (90) (7.47%), chrysoeriol (93) (3.55%) and luteolin (87) (3.33%) (Appendix A).
ijms-24-09999-t005_Table 5Table 5Secondary metabolites in *Echinops ritro* extracts.No.Identified/Tentatively Annotated CompoundMolecular FormulaExact Mass[M-H]^−^t_R_(min)Δ ppmDistribution**Hydroxybenzoic acids, hydroxycinnamic acids, phenylethanoid glycosides and coumarins****1**protocatechuic acid-*O*-hexosideC_13_H_16_O_9_315.07271.671.316A, B**2**vanillic acid-*O*-hexoside ^b^C_14_H_18_O_9_329.08751.76−0.381A, B**3**protocatechuic acid ^a^C_7_H_6_O_4_153.01812.01−8.247A, B**4**vanillyl alcohol-hexoside ^b^C_14_H_20_O_8_315.10852.121.299A, B**5***p*-hydroxyphenylacetic acid *O*-hexoside ^b^C_14_H_18_O_8_313.09292.133.926A, B**6**syringic acid-*O*-hexoside ^b^C_15_H_20_O_10_359.09852.290.473A, B**7**hydroxybenzoic acid-*O*-hexoside ^b^C_13_H_16_O_8_299.07782.430.800A, B**8***p*-hydroxyphenylacetic acid ^b^C_8_H_8_O_3_151.04012.50−8.789A, B**9**gentisic acid-*O*-hexoside ^b^C_13_H_16_O_9_315.07272.551.316A, B**10**syringic acid-*O*-pentoside ^b^C_14_H_18_O_9_329.08782.691.199A, B**11**aesculetin-*O*-hexoside ^b^C_15_H_15_O_9_339.07242.710.515A, B**12**syringyl-*O*-hexose ^b^C_15_H_20_O_10_359.09842.762.952A, B**13**4-hydroxybenzoic acid ^a,b^C_7_H_6_O_3_137.02302.84−10.928A, B**14**hydroxybenzoic acid-*O*-hexoside isomer ^b^C_13_H_16_O_8_299.07783.00−2.443A, B**15***p*-hydroxyphenylacetic acid *O*-hexoside isomer ^b^C_14_H_18_O_8_313.09293.010.030A, B**16***O*-caffeoyl hexose ^b^C_15_H_18_O_9_341.08713.08−2.068A, B**17**quinic acid ^b^C_7_H_12_O_6_191.05493.15−6.079A, B**18***p*-hydroxyphenylacetic acid *O*-hexoside isomer ^b^C_14_H_18_O_8_313.09293.300.030B**19**coumaric acid-*O-*hexoside ^b^C_15_H_18_O_8_325.09303.33−1.386A, B**20***p-*coumaric acid ^a,b^C_9_H_8_O_3_163.03893.35−6.547A, B**21**shikimic acid ^b^C_7_H_10_O_5_173.04553.37−5.875A, B**22**aesculetin ^b^C_9_H_6_O_4_177.01933.46−6.225A, B**23**vanillyl alcohol- (acetyl)-hexoside ^b^C_16_H_22_O_9_357.11913.473.681A, B**24**caffeic acid ^a,b^C_9_H_8_O_4_179.03393.55−6.211A, B**25**hydroxybenzoic acid-dihexoside ^b^C_19_H_26_O_13_461.13013.63−0.399A, B**26**ferulic acid ^a,b^C_10_H_10_O_4_193.04943.79−6.330A, B**27**gentisic acid ^a,b^C_7_H_6_O_4_153.01803.86−8.443A, B**28***o*-coumaric acid ^a,b^C_9_H_8_O_3_163.03894.56−7.160A**29**vanillic acid ^a^C_8_H_8_O_4_167.03384.78−7.376A**30**caffeic acid-(salicyl)-dihexoside ^b^C_28_H_32_O_16_623.16185.781.833A, B**31**gentisic acid-(caffeoyl)-hexoside ^b^C_22_H_22_O_12_477.10386.53−2.996A, B**32**protocatechuic acid-(salicyl)-hexoside ^b^C_20_H_20_O_11_435.09337.651.208A, B**Mono-, diacyl- and triacylquinic acids and their hexosides****33**1-caffeoylquinic acid ^b^C_16_H_18_O_9_353.08761.89−0.893A, B**34**neochlorogenic (3-caffeoylquinic) acid ^a^C_16_H_18_O_9_353.08672.36−0.100A, B**35**caffeoylquinic acid-hexoside ^b^C_22_H_27_O_14_515.13952.861.420A, B**36**3-*p*-coumaroylquinic acid ^b^C_16_H_18_O_8_337.09283.02−2.524A, B**37**chlorogenic (5-caffeoylquinic) acid ^a^C_16_H_18_O_9_353.08743.19−1.233A, B**38**4-caffeoylquinic acid ^b^C_16_H_18_O_9_353.08783.37−0.100A, B**39**3-feruloylquinic acid ^b^C_17_H_20_O_9_367.10403.441.402A**40**caffeoylquinic acid-hexoside ^b^C_22_H_27_O_14_515.13953.741.420A, B**41**5-caffeoylquinic acid isomerC_16_H_18_O_9_353.08743.891.543A, B**42**5-*p*-coumaroylquinic acid ^b^C_16_H_18_O_8_337.09283.95−0.180A, B**43**3-caffeoyl-5-hydroxy-dihydrocaffeoylquinic acid ^b^C_25_H_26_O_13_533.12884.021.043A, B**44**4-hydroxy-dihydrocaffeoyl-5-caffeoylquinic acid ^b^C_25_H_26_O_13_533.12884.244.232A, B**45**5-feruloylquinic acid ^b^C_17_H_20_O_9_367.10344.41−0.096A, B**46**1-caffeoyl-3-hydroxy-dihydrocaffeoylquinic acid ^b^C_25_H_26_O_13_533.12884.431.043A, B**47**4-feruloylquinic acid ^b^C_17_H_20_O_9_367.10344.661.157A, B**48**dicaffeoylquinic acid-hexosideC_31_H_34_O_17_677.17235.172.492A, B**49**dicaffeoylquinic acid-hexoside isomerC_31_H_34_O_17_677.17235.252.580A, B**50**1-hydroxy-dihydrocaffeoyl-3-caffeoylquinic acid ^b^C_25_H_26_O_13_533.12885.061.831A, B**51**dicaffeoylquinic acid-hexoside isomer ^b^C_31_H_34_O_17_677.17235.662.403B**52**3,4-dicaffeoylquinic acid ^a,b^C_25_H_24_O_12_515.11905.691.321A, B**53**3,5-dicaffeoylquinic acid ^a,b^C_25_H_24_O_12_515.11895.852.020A, B**54**3-caffeoyl-5-dehydrocaffeoylquinic acid ^b^C_25_H_22_O_12_513.10185.87−3.974A, B**55**1,5-dicaffeoylquinic acid ^a,b^C_25_H_24_O_12_515.11906.030.720A, B**56**4,5-dicaffeoylquinic acid ^a,b^C_25_H_24_O_12_515.11906.22−1.163A, B**57**3-caffeoyl-5-*p*-coumaroylquinic acid ^b^C_25_H_24_O_11_499.12516.531.513A, B**58**4-*p*-coumaroyl-5-caffeoylquinic acid ^b^C_25_H_24_O_11_499.12526.92−0.630A, B**59**4-feruloyl-5-caffeoylquinic acid ^b^C_26_H_26_O_12_529.13567.09−0.490A, B**60**3,4,5-tricaffeoylquinic acid ^a,b^C_34_H_30_O_15_677.15287.782.417A, B**Flavonoids****61**kaempferol *O*-dihexoside ^b^C_27_H_30_O_16_609.14644.45−1.629A, B**62**eriodyctiol *O*-dihexoside ^b^C_27_H_32_O_16_611.16184.65−0.324A, B**63**rutin ^a,b^C_27_H_30_O_16_609.14645.092.121A, B**64**isoquercitrin ^a^C_21_H_20_O_12_463.08855.150.585A, B**65**luteolin 7-*O*-rutinoside ^a,b^C_27_H_30_O_15_593.15125.221.832A, B**66**patuletin *O*-rutinoside ^b^C_28_H_32_O_17_639.15675.222.484B**67**hyperoside ^a^C_21_H_20_O_12_463.08855.290.585A, B**68**luteolin-*O*-hexuronide ^b^C_21_H_18_O_12_461.07365.370.718A, B**69**luteolin-7-*O*-glucoside ^a,b^C_21_H_19_O_11_447.09345.390.348A, B**70**patuletin *O*-hexoside ^b^C_22_H_22_O_13_493.09875.451.655A, B**71**quercetin *O*-acetylhexoside ^b^C_23_H_22_O_13_505.09885.602.210A, B**72**kaempferol 3-*O*-rutinoside ^a,b^C_27_H_30_O_15_593.15125.641.630A, B**73**isorhamnetin 3-*O*-rutinoside ^a,b^C_28_H_32_O_17_623.16185.782.025A, B**74**apigenin *O*-rutinoside ^b^C_27_H_30_O_14_577.15635.811.960A, B**75**kaempferol-3-*O*-glucoside ^a,b^C_21_H_19_O_11_447.09345.860.348A, B**76**isorhamnetin 3-*O*-glucoside ^a,b^C_22_H_21_O_12_477.10446.011.092A, B**77**apigenin 7-*O*-glucoside ^a,b^C_21_H_19_O_10_431.09806.07−0.881A, B**78**apigenin *O*-hexuronideC_21_H_18_O_11_445.07746.12−0.617A, B**79**kaempferol *O*-acetylhexoside ^b^C_23_H_22_O_12_489.10386.291.433A, B**80**chrysoeriol *O*-hexoside ^b^C_22_H_22_O_11_461.10936.310.684A, B**81**chrysoeriol *O*-hexuronide ^b^C_22_H_20_O_12_475.08856.330.381A, B**82**isorhamnetin *O*-acetylhexoside ^b^C_24_H_24_O_13_519.11446.471.707B**83**jaceosidine O-hexoside ^b^C_23_H_24_O_12_491.11956.501.875A, B**84**eupatorin *O*-hexoside ^b^C_24_H_26_O_12_505.13516.672.139A, B**85**cirsimaritin *O*-hexoside ^b^C_23_H_24_O_11_475.12466.761.379A, B**86**luteolin *O*-acetylhexoside ^b^C_23_H_22_O_12_489.10386.852.312B**87**luteolin ^a,b^C_15_H_9_O_7_285.04067.580.346A, B**88**patuletin ^b^C_16_H_12_O_8_331.04647.701.388B**89**apigenin O-acetylhexoside ^b^C_23_H_22_O_11_473.10897.821.406A, B**90**apigenin ^a,b^C_15_H_9_O_5_269.04558.61−0.285A, B**91**hispidulin ^b^C_16_H_12_O_6_299.05618.840.430A, B**92**cirsiliol ^b^C_17_H_14_O_7_329.06778.871.045A, B**93**chrysoeriol ^a,b^C_16_H_12_O_6_299.05618.920.330A, B**94**spinacetin ^b^C_17_H_14_O_8_345.06168.990.897A, B**95**apigenin-*O*-(*p*-coumaroyl)hexosideC_30_H_26_O_12_577.13419.021.665A, B**96**apigenin-*O*-(*p*-coumaroylhexoside) isomerC_30_H_26_O_12_577.13419.071.872A, B**97**quercetagetin-3,6,3′(4′)-trimethyl ether ^b^C_18_H_16_O_8_359.07729.071.196B**98**isorhamnetin ^a,b^C_16_H_12_O_7_315.05129.110.870A, B**99**jaceosidin (6-hydroxyluteolin-6,3′-dimethyl ether) ^a,b^C_17_H_14_O_7_329.06779.141.319A, B**100**naringenin *O*-(*p*-coumaroyl)hexosideC_30_H_28_O_12_579.14979.151.241A, B**101**naringenin *O*-(*p*-coumaroylhexoside) isomerC_30_H_28_O_12_579.14979.250.871A, B**102**axillarin ^b^C_17_H_14_O_8_345.06169.251.042A, B**103**quercetin-7,3′(4′)-dimethyl ether ^b^C_17_H_14_O_7_329.06779.441.319A, B**104**cirsimaritin ^b^C_17_H_14_O_6_313.071910.381.401A, B**105**eupatilin ^a,b^C_18_H_16_O_7_343.081210.670.653A, B**106**ayanin ^b^C_18_H_16_O_7_343.081211.030.390A, B**107**genkwanin ^a,b^C_16_H_12_O_5_283.061211.59−0.103A, B**108**ayanin isobar ^b^C_18_H_16_O_7_343.081211.760.653A, B^a^ Identified by comparison with an authentic standard; ^b^ reported for the first time in *E. ritro*; A—flower head extract; B—leaf extract.


## 3. Discussion

Oxidative stress is a major pathophysiological process in many liver diseases. However, the application of antioxidants from various natural sources shows significant clinical potential. Oxidative stress can be both a cause of hepatocyte damage, dysfunction and cell death and a consequence of liver injury. This dualism requires a clarification of the causal relationship via experimental studies of the hepatotoxic mechanisms of pro-oxidants, including drugs, and specific prophylactic and therapeutic approaches via the administration of antioxidants of plant origin. Experimental models using different biomarkers to assess hepatotoxicity should evaluate the possible mechanisms of injury as well as the possible hepatoprotective potential of plant extracts for humans. This approach will increase the potential for liver protection established in cell culture and animal models to be successfully extrapolated to human pathophysiology and therapy.

The production of ROS has been proposed as an early event of drug-induced hepatotoxicity and as an indicator of hepatotoxic potential of medications [33]. It has been discovered that many drugs can induce oxidative stress including an increase in cellular oxidants and lipid peroxidation, the depletion of antioxidants in the liver, a decrease in the activity of antioxidant enzymes, etc. [34].

In addition to liver disease, oxidative stress is an important pathophysiological mechanism in numerous other socially relevant chronic diseases. Many biologically active substances and secondary metabolites of plant origin, mainly phenolic compounds, are being investigated in experimental pharmacology as a source of antioxidants with a crucial role in the prevention of oxidative-stress-related diseases. Flavonoids have been found to have the potential to break free radical chains in lipids (by donating hydrogen atoms to lipids or lipid peroxyl radicals) [35]. Silymarin, a mixture of polyphenolic flavonoids derived from the fruits and seeds of *Silybum marianum* (milk thistle) [36], is one of the most commonly used hepatoprotective and antioxidant drugs in medical practice. However, there are not many other antioxidants with therapeutic application in the clinic. Therefore, the identification of new sources of compounds with potent antioxidant and hepatoprotective activity is considered a crucial step in the prevention and treatment of drug-induced liver injury. 

The aim of this study was to elucidate the possible mechanisms of the antioxidant and hepatoprotective effects of *E. ritro.* The leaf (ERLE) and flowering head (ERFE) extracts of the species demonstrated a good antioxidant potential in DPPH, ABTS, FRAP, CUPRAC, metal chelating and phosphomolybdenum inhibitory tests (Table 1). Both extracts possess high antioxidant potential, especially the ERLE. In contrast to these spectrophotometric direct methods in tests with biological structures (subcellular fractions and primary isolated hepatocytes), the ERFE demonstrated a better effect (Figure 1, Figure 2 and Figure 3). We do not consider this to be a contradiction as we demonstrated an antioxidant effect via different mechanisms, which shows the great potential of this plant species. The antioxidant properties of the species have also been investigated by Zengin et al. [24] and Aydin et al., [17]. As the ERFE showed better hepatoprotective and antioxidant effects compared to the ERLE in Fe/AA-induced oxidative stress in liver microsomes (Figure 1) and in t-BuOOH- induced hepatocytes damage (Figure 2, Figure 3 and Figure 4), this extract was used for further in vivo experiments with diclofenac (DF)-induced hepatic oxidative stress and toxicity in male Wistar rats. 

ERFE acts as an antioxidant by increasing the GSH level in hepatocytes and the activity of antioxidant enzymes GR GPx, and GST, which are involved in GSH turnover (Figure 3, Table 3). It also decreased the formation of MDA, a marker of lipid peroxidation. When it is administered in combination with DF, the ERFE decreased its hepatotoxicity by improving antioxidant defense status in the experimental rats. Therefore, the apparent hepatoprotective effect might be due to the ability of the ERFE to neutralize the increase in free radicals caused by diclofenac. In our investigation, we proved the pro-oxidant effect of DF via the increased quantity of liver MDA, while the activities of antioxidant enzymes GR, GPx and GST and the level of GSH were remarkably reduced (Table 3). The decreased activity of these enzymes may significantly contribute to diclofenac cytotoxicity. These enzymes may increase the efficiency of GSH to neutralize the reactive metabolites of diclofenac. GST play an important role in the inactivation of DF quinone imines, especially in stress conditions when tissue levels of GSH are decreased [37], as found in the present experiment. Thus, GSH depletion can be considered one of the main biomarkers of diclofenac-induced oxidative liver injury. GSH deficiency has been shown to induce the binding of endogenous reactive oxygen species (ROS) to cellular macromolecules, which in turn activates lipid peroxidation processes, which we found with the increased MDA levels in the livers of diclofenac-treated rats [38].

The formation of reactive intermediate products via oxidative CYP-mediated metabolism is considered to play a role in the development of DF-induced hepatotoxicity. It was found that two CYP isoforms were involved in the bioactivation of DF, namely, CYP3A4 and CYP2C9 [3]. In the present study, DF significantly increased the activity of the two enzymes we examined, EMND, a marker of CYP3A4 activity [39], and AH, a marker of CYP2E1 activity [40] (Table 4). Although there is no evidence that CYP2E1 is involved in the bioactivation of diclofenac and the formation of toxic quinone imines, increased microsomal EMND activity, as established in the present study, could increase the risk of liver toxicity. Reducing the expression and activity of each one of these isoforms will lead to the reduced toxicity of diclofenac, as we confirmed in this study. We observed that, either administered alone or with diclofenac, the ERFE decreased the activity of EMND. According to [41], this is probably due to the direct inactivation or inhibition of enzyme expression. 

Furthermore, we suggested that the ERFE might exert its hepatoprotective effects via the modulation of hepatic biotransformation. Several studies have found polyphenols to interact with cellular defense systems such as phase I (mainly the CYP450 complex enzymes) and II (e.g., glutathione transferases and glucuronyl transferases) detoxifying enzymes [42]. It is well known that plant sources containing flavonoids (e.g., ERFE) have membrane-stabilizing activities and hepatoprotective, antioxidant and CYP-inhibiting effects [41].

The ERFE, administered alone at three different doses in the acute toxicity evaluation test, produced no toxic effects in the experimental animals, which was also established by the subsequent histopathological study, proving its safety.

At the end of the present experiment, ASAT and GGT activities and triglyceride and bilirubin levels in the serum of DF-treated rats were higher than in the control rats but remained within the reference range (Table 2). We supposed that the dose of 50 mg/kg is probably sufficient to cause oxidative stress in the liver cells but does not cause serious and irreversible damage to the hepatocyte membranes and a corresponding increase in serum biochemical markers.

Protocatechuic acid derivatives are well known for their antioxidant and hepatoprotective activity [43]. Protocatechuic acid alone promoted endogenous and antioxidant enzymatic activities and inhibited ROS generation in a rat model of oxidative stress injury. 

Quinic acid, another major compound in *E. ritro* leaves, exerted pronounced dose-dependent antioxidant activity in the cell model of H_2_O_2_-induced oxidative stress, restoring the MDA levels [44].

3,5- and 4,5-dicaffeoylquinic and chlorogenic acid possessed strong antioxidant effects [45]. Caffeoylquinic acids (CQAs) exhibited a radical scavenging activity similar to that of ascorbic acid. They chelated transition metals (Fe^2+^ and Cu^2+^) and disrupted chain reactions. Chlorogenic acid can scavenge different radicals and protect DNA from damage caused by oxidative stress. 3,5-dicaffeoylquinic acid (isolated from *Geigeria alata*) ameliorated the oxidative stress biomarkers GSH and MDA and serum parameters [46]. The protective effects on the antioxidant enzymes were evidenced by the restored GR and GST activity in the liver of diabetic rats. 

Apigenin is a very common flavonoid in *Echinops* species [11]. From a pharmacological standpoint, significant advances have been made in understanding the biochemical mechanisms underlying apigenin effects in liver injuries and diseases [47]. In addition to evoking a GSH- and GR-dependent antioxidant response, apigenin also displayed inhibitory activity on lipid accumulation and the excessive production of pro-inflammatory cytokines in the liver. 

Apigenin 7-O-glucoside is known to exert stronger inhibition against free-radical-induced damage than Trolox [48] Both apigenin and apigenin 7-O-glucoside restrained the LPS-induced NLRP3/NF-κB signaling pathway. Apigenin 7-O-glucuronide has already been shown to be present in *E. ritro* [24]. It is known to exert antioxidant and anti-inflammatory activity [49]

Kaempferol 3-O-glucoside and kaempferol 3-O-rutinoside showed almost equal efficiency in DPPH and ABTS radical scavenging [50].

Among quercetin-3-O-glycosides (hyperoside, rutin and isoquercitrin), quercetin-3-O-β-D-galactoside (hyperoside) exhibited a slightly higher DPPH value than quercetin-3-O-β-D-glucoside (isoquercitrin). Both flavonoids have adjacent phenolic hydroxyl groups at the B-ring, which is responsible for their antioxidant activity reported in the literature [51]. 

As highlighted in a recent review, chrysoeriol scavenged intracellular ROS generation and exerted dose-dependent antioxidant activity in the Fe^2+^/ascorbate-induced lipid peroxidation model in rat liver [52]. Chrysoeriol and chrysoeriol 7-O-glucopyranoside alleviated peroxidative damage in rat liver, as evidenced by the decrease in liver and blood levels of enzymic and non-enzymic antioxidants. Cirsiliol reduced the free radical production by the mammalian retinal rod outer segments [53]. The authors reported a significant protective effect of cirsiliol on the structural stability of rod OS, suggesting that it may be considered a promising compound against oxidative stress.

Herein, a comprehensive UHPLC-HRMS analysis of leaves and flowering heads was performed, yielding the identification/annotation of 108 secondary metabolites. For the first time, 29 phenolic acids and derivatives together with coumarins, 25 acylquinic acids and 41 flavonoid aglycones and glycosides were reported in *E. ritro*. To the best of our knowledge, phenylethanoid glycosides, hydroxycinnamoyl hexoses (sugar esters), a series of hydroxybenzoic and hydroxicinnamic acids glycosides; coumaroyl- and feruloylquinic acids, caffeoyl-hydroxydihydrocaffeoylquinic acids, caffeoyl-feruloylquinic and caffeoyl-dehydrocaffeoylquinic acids, tricaffeoylquinic acid, and dicaffeoylquinic acid and hexosides are reported for the first time. Moreover, except for mono- and dicaffeoylquinic acids, the aforementioned acylquinic subclasses even include new compounds for the *Echinops* genus. In the same manner, flavonoid rutinosides, acetylhexosides, methoxylated flavonoids and their glycosides represent new secondary metabolites in the species.

## 4. Materials and Methods

### 4.1. Plant Material

The *E. ritro* leaves and flowering heads were collected from Karjali, Bulgaria during the full flowering stage in July 2021. The plant was identified by one of us (D.Z.) as recommended by Stojanov (2012) [22]. Voucher specimen was deposited at Herbarium Facultatis Pharmaceuticae Sophiensis (voucher specimen № 11620). Twenty plant samples were separated into leaves and flowering heads and dried at room temperature. 

### 4.2. Chemicals

Acetonitrile (hypergrade for LC-MS), formic acid (for LC-MS) and methanol (analytical grade) were purchased from Merck, Sofia, Bulgaria. The authentic standards used for compound identification were obtained from Extrasynthese (Genay, France) for protocatechuic, p-hydroxyphenylacetic, vanillic, gentisic, ferulic, p-coumaric, o-coumaric acid, rutin, hyperoside, luteolin 7-O-rutinoside, luteolin 7-O-glucoside, isoquercitrin, kaempferol 3-O-rutinoside, isorhamnetin 3-O-rutinoside, kaempferol 3-O-glucoside, isorhamnetin 3-O-glucoside, apigenin 7-O-glucoside, quercetin, apigenin, luteolin, chrysoeriolq eupatilin and genkwanin. Caffeic, 3,4-dicaffeoylquinic, 3,5-dicaffeoylquinic 1,5-dicaffeoylquinic, 4,5-dicaffeoylquinic and 3,4,5- tricaffeoylquinic acids were supplied by Phytolab (Vestenbergsgreuth, Germany). Chlorogenic acid and isorhamnetin were purchased from Sigma-Aldrich (St. Louis, MO, USA). Diclofenac, silymarin, collagenase, 1-chloro-2,4-dinitrobenzene, beta-nicotinamide adenine dinucleotide 2′-phosphate reduced tetrasodium salt (NADPH), ethylenediaminetetraacetic acid (EDTA), bovine serum albumin (fraction V) and 2,2′-dinitro-5,5′dithiodibenzoic acid (DTNB) were obtained from Merck (Darmstadt, Germany). Reduced glutathione (GSH), oxidized glutathione (GSSG), glutathionereductase (GR) and cumene hydroperoxide were purchased from Sigma Chemical Co. (Taufkirchen, Germany). 

### 4.3. Sample Extraction

Air-dried powdered leaves and flowering heads (100 g) were extracted twice with 80% MeOH (1:20 *w*/*v*) by sonication (80 kHz, ultrasound bath Biobase UC-20C) for 15 min at room temperature. The extracts were concentrated in vacuo and lyophilized (lyophilizer Biobase BK-FD10P) to yield crude extracts of 5.10 g (leaves) and 4.51 g (flowering heads). The lyophilized extracts were dissolved in 80% methanol (0.1 mg/mL). An aliquot (2 mL) of each extract solution was filtered through a 0.45 μm syringe filter (Polypure II, Alltech, Lokeren, Belgium) and subjected to UHPLC–HRMS analyses.

### 4.4. Determination of Antioxidant Effects

Intrinsic scavenging/reducing properties of the extracts (0.2–1 mg/mL) were determined via colorimetric assays [26].

### 4.5. In Vitro Studies on Antioxidant Activity and Hepatotoxicity

The in vitro experiments were carried out in freshly isolated rat liver microsomes and hepatocytes. Cell viability, LDH leakage into the medium, GSH levels and MDA quantity were assessed.

#### 4.5.1. Isolation of Rat Liver Microsomes

Liver microsomes from 5 rats were isolated and stored at −70 °C until use [54]. The content of microsomal protein was determined according to the method of Lowry, using bovine serum albumin as a standard [55].

On the day of the experiment, microsomes were resuspended in 0.1 M potassium phosphate buffer (1 mg/mL) and preincubated for 15 min at 37 °C with two different concentrations, a low concentration (LC, 10 µg/mL) and a high concentration (HC, 50 µg/mL), of ERFE and ERLE. In order to induce lipid peroxidation, 20 μM FeSO_4_ and 500 μM Ascorbic acid were added directly into abovementioned suspensions and incubated for an additional 30 min at 37 °C. Then, MDA was assessed and measured using a molar extinction coefficient of 1.56 × 10^5^ M^−1^cm^−1^ [56].

#### 4.5.2. Isolation and Incubation of Primary Rat Hepatocytes

Five rats were anesthetized with sodium pentobarbital (0.2 mL/100 g). In situ liver perfusion and cell isolation were performed as described by Fau et al. [57] with modifications [58]. Cells were counted under the microscope, and the viability was assessed by trypan blue exclusion (0.05%) [57].

Subsequently, the hepatocytes were incubated for 30 min with the same concentrations (LC and HC) of ERFE and ERLE used in the previous experiment with microsomes. For the induction of oxidative stress, cells were incubated with 75 μM t-BuOOH for 1 h [59]. GSH content and MDA level in hepatocytes were measured according to Fau et al. [57]. Lactate dehydrogenase release from the hepatocytes was measured spectrophotometrically at 314 nm by using commercially available LDH kit.

### 4.6. In Vivo Experiments

#### 4.6.1. Animals

The Animal Care Ethics Committee approved the study protocol and issued an ethical clearance (No. 346 of 28 February 2023) from the Bulgarian Food Safety Agency. The rats were housed, maintained and euthanized in accordance with the relevant international rules and recommendations as outlined in the European Convention for the Protection of Vertebrate Animals used for Experimental and other Scientific Purposes (ETS 123).

Forty-three specific pathogen-free male Wistar rats at three months of age (150–200 g) were used. Rats were housed in Plexiglas cages (three per cage) in a 12/12 light/dark cycle under standard laboratory conditions (ambient temperature 20 ± 2 °C and humidity 72 ± 4%). All animals were purchased from the National Breeding Center, Sofia, Bulgaria and allowed a minimum of 7 days to acclimate to the new conditions before the start of the study. Standard complete commercial pelleted chow suitable for their strain and age and fresh drinking water were available ad libitum throughout the experimental period of 28 days.

#### 4.6.2. Acute Toxicity Test in Rats

Acute toxicity was assessed in 9 male Wistar rats after oral (p.o.) administration of the ERFE using the simplified method described by Lorke [60] with slight modifications. Three animals were used per dose at 3 fixed-dose intervals, with 3000, 2000, and 1000 mg/kg. Animals were observed every 3 h for the first 24 h and once a day for up to 14 days. During this period, the behavior and the basic activities of animals were observed. On day 14, they were euthanized by decapitation with laboratory guillotine after anesthesia with ketamine/xylazine (80/10 mg/kg, i.p.), and an examination of the internal organs for possible macroscopic abnormalities was conducted. Small pieces of liver from all rats were preserved in 10% formalin for histopathological assessment.

Histopathological studies

For light microscopic evaluation, liver tissues were fixed in 10% buffered formalin, and, then, thin sections (4 μm) were subsequently stained with hematoxylin/eosin to determine general histological features [61]. Sections were studied under light microscope Leica DM 500.

#### 4.6.3. Experimental Design for Evaluation of Protective Effects of ERFE on Diclofenac-Induced Oxidative Liver Damage

Twenty-four male Wistar rats were randomized into four experimental groups with six animals in each (n = 6). DF and ERFE were administered as follows:

Group 1 included control animals treated with physiological saline via oral gavage (0.5 mL/g bw). Group 2 included animals orally exposed to ERFE alone (1/10 of LD_50_ or 300 mg/kg/day), which was dissolved in a physiological saline buffer for 14 days. Group 3 received orally DF sodium (50 mg/kg) for 3 consecutive days (from day 5 to day 7). Group 4 was exposed orally to ERFE for 14 days, receiving p.o. DF sodium (50 mg/kg) on days 5, 6 and 7, 20 min after ERFE administration.

On the 15th day, the animals were sacrificed with a laboratory guillotine, blood was collected, and biochemical parameters in serum were measured. Liver parts were taken to study the biomarkers of oxidative stress, malondialdehyde (MDA) and glutathione (GSH), as well as the activity of the antioxidant enzymes glutathione peroxidase (GPx), glutathione reductase (GR) and glutathione-S-transferase (GST). The activities of AH and EMND, which are markers of CYP2E1 and CYP3A4 activity, respectively, were also evaluated. For all subsequent studies, excised organs were perfused with cold saline (0.9% NaCl), dried, weighed and homogenized with appropriate buffers.

##### Assessment of Serum Biochemical Parameters

Biochemical markers including glucose, urea, creatinine, total protein, albumin, total bilirubin, direct bilirubin, transaminases L-aspartate aminotransferase (ASAT) and L-alanine aminotransferase (ALAT), alkaline phosphatase, gamma-glutamyltransferase (gGT), amylase, total cholesterol, triglycerides and uric acid were assessed using commercial kits for biochemical analyzer “Mindray BS-120” as described by the manufacturer. 

##### Measurement of Lipid Peroxidation (LPO) in Liver Homogenate

Lipid peroxidation was determined by measuring the formation rate of thiobarbituric acid (TBARS) (expressed as malondialdehyde (MDA) equivalents) as described by Polizio and Pena [62]. The amount of MDA was calculated as described for microsomes and hepatocytes and expressed in nmol/g wet tissue.

##### Measurement of Reduced Glutathione (GSH) in Liver Homogenate

GSH was estimated by measuring non-protein sulfhydryls after protein precipitation with TCA using the method described by Bump et al. [63]. Absorbance was determined as described for hepatocites, and results were expressed in nmol/g of wet tissue.

##### Preparation of Liver Homogenates for Antioxidant Enzyme Activity Measurement

The livers were rinsed in ice-cold saline and finely sieved. A total of 10% homogenates were prepared in 0.05 M phosphate buffer (pH = 7.4) and centrifuged at 7000× *g*, and the supernatants were used for antioxidant enzyme assay. The protein content of the liver homogenates was measured by the method of Lowry (1951). 

Gluthatione peroxidase activity (GPx) was measured by NADPH oxidation, using a coupled reaction system consisting of reduced glutathione and glutathione reductase [64]. The reaction was initiated by adding 50 µL cumene hydroperoxide (1 mg/mL), and the rate of disappearance of NADPH with time was determined by monitoring absorbance at 340 nm. Results are expressed in nmol/mg protein/min. 

Glutathione reductase avtivity (GR) was measured according to the method of Pinto et al. [65] by following NADPH oxidation spectrophotometrically at 340 nm and using an extinction coefficient of 6.22 mM^−1^ cm^−1^. 

Glutathione-S-transferase activity (GST) was measured using 1-chloro-2,4-dinitrobenzene (CDNB) as a substrate [66]. The enzyme activity is expressed as nmol of CDNB-GSH conjugate formed/min/mg protein. 

##### Aniline 4-Hydroxilase Activity (AHA)

For evaluation of phase I enzyme activity, liver microsomes, from all experimental groups, were prepared as described above.

The method for assessment of AHA used 4-hydroxilation of aniline and 4-aminophenol, which are chemically converted to a phenol–indophenol complex with an absorption maximum at 630 nm. Enzyme activity was expressed as nmol/min/mg protein [67].

##### Ethylmorphine-N-demethylase Activity (EMNDA)

The enzyme activity of CYP3A4 was evaluated by the formation of formaldehyde, trapped in the solution as semicarbazone and measured by the colorimetric procedure at 415 nm. Enzyme activity was expressed as nmol/min/mg protein [67].

#### 4.6.4. UHPLC-HRMS

Mass spectrometry analyses were carried out on a Q Exactive Plus mass spectrometer (ThermoFisher Scientific, Inc., Waltham, MA, USA) equipped with a heated electrospray ionization (HESI-II) probe. The apparatus was operated in negative and positive modes within the *m/z* range from 100 to 1000. The chromatographic separation was performed on a reversed phase column Kromasil EternityXT C18 (Göteborg, Sweden) (1.8 µm, 2.1 × 100 mm) at 40 °C. The chromatographic analyses were performed as previously described [26]. MZmine 2 software was applied to the UHPLC–HRMS raw files of the studied *E. ritro* extracts for the semi-quantitative analysis. Results are expressed as % of peak area of the compound to the total peak areas of the corresponding group’s secondary metabolites.

#### 4.6.5. Statistical Analysis

Statistical program ‘MEDCALC’ was used for analysis of the data. For the in vitro experiments, the results are expressed as mean ± SEM for five experiments with three parallel samples. The data from in vivo experiments are expressed as mean ± SEM of four rats in each group. The significance of the data was assessed using the nonparametric Mann–Whitney U test. Values of *p* ≤ 0.05 were considered statistically significant.

## 5. Conclusions

The results obtained from the in vitro and in vivo experiments revealed that ERFE does not exhibit toxic effects and that it is safe at the tested dose. It reduces oxidative stress, successfully restores liver function and ameliorates diclofenac-induced liver injury. The ERFE protective mechanisms could have the potential to increase the activity of antioxidant enzymes and the amount of endogenous antioxidant GSH in in vitro/in vivo conditions. Furthermore, the ERFE beneficial effect could also be associated with its metabolite-mediated activities and its inhibitory role in diclofenac bioactivation and toxic metabolite formation. A variety of hydroxybenzoic, hydroxycinnamic and acylquinic acids; phenylethanoid glycosides; and flavonoids in the flowering head and leaf extracts could be related to their antioxidant and hepatoprotective activity. As rich sources of natural compounds, *E. ritro* extracts could promote health benefits and provide pharmaceutical applications. 

## Figures and Tables

**Figure 1 ijms-24-09999-f001:**
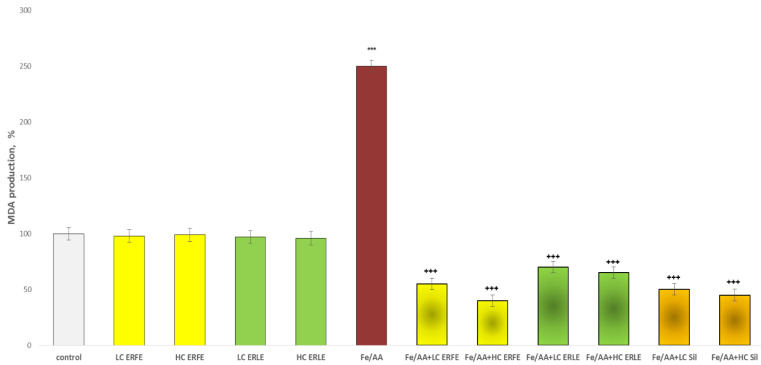
Effects of ERFE and ERLE at LC 10 µg/mL and HC 50 µg/mL, administered alone and in conditions of non-enzyme (Fe/AA-induced) lipid peroxidation, on MDA production in isolated rat liver microsomes. *** *p* < 0.001 vs. control (non-treated rat liver microsomes); ^+++^ *p* < 0.001 vs. Fe/AA.

**Figure 2 ijms-24-09999-f002:**
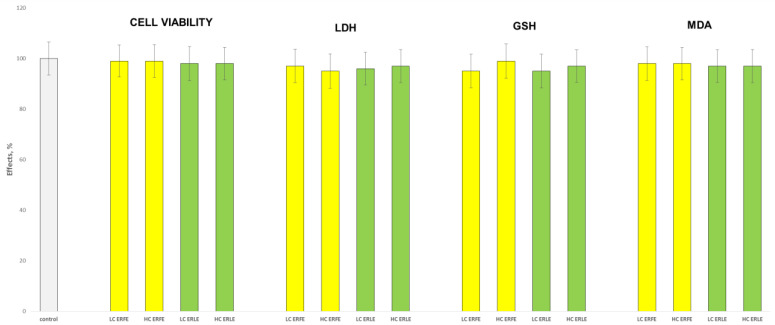
Effects of ERFE and ERLE at LC and HC, administered alone, on the main parameters, characterized by the functional and metabolic status of isolated rat hepatocytes.

**Figure 3 ijms-24-09999-f003:**
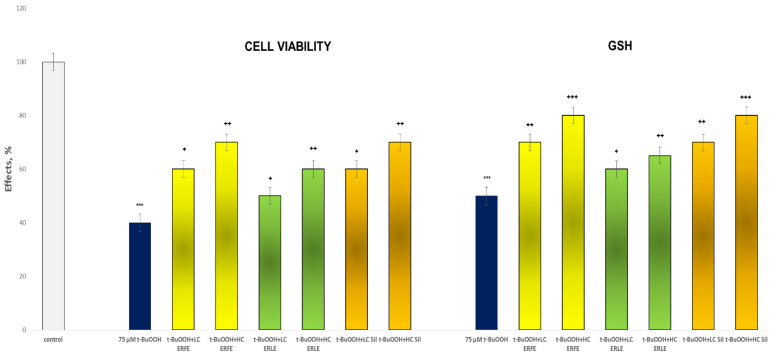
Effects of ERFE and ERLE at LC and HC, in conditions of t-BuOOH-induced oxidative stress, on cell viability and GSH level in isolated rat hepatocytes. *** *p* < 0.001 vs. control (non-treated rat hepatocytes); ^+^ *p* < 0.05; ^++^ *p* < 0.01; ^+++^ *p* < 0.001 vs. t-BuOOH.

**Figure 4 ijms-24-09999-f004:**
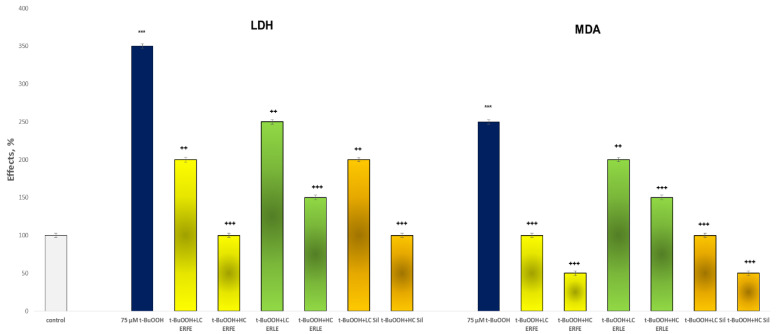
Effects of ERFE and ERLE at LC and HC, in conditions of t-BuOOH-induced oxidative stress, on LDH leakage and MDA production in isolated rat hepatocytes. *** *p* < 0.001 vs. control (non-treated rat hepatocytes); ^++^ *p* < 0.01; ^+++^ *p* < 0.001 vs. t-BuOOH.

**Figure 5 ijms-24-09999-f005:**
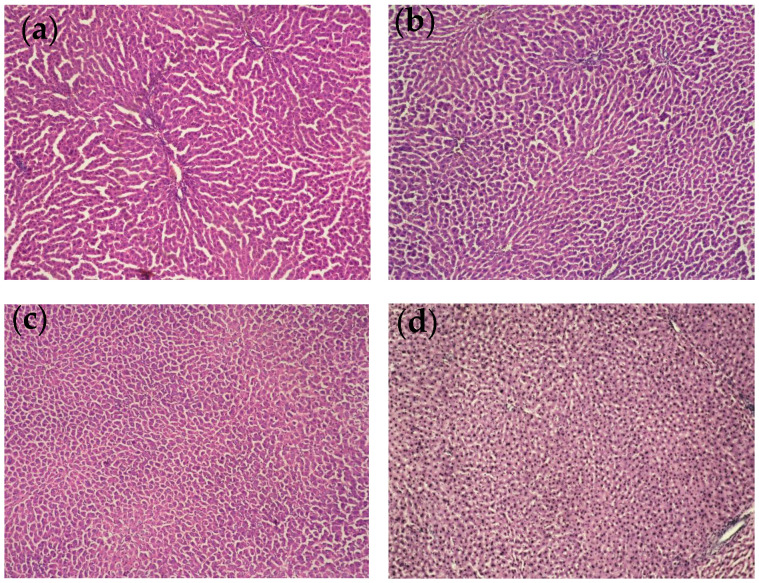
Histological profile. Liver tissues from all treated groups revealed normal cellular architecture without pathological changes. (**a**) Control rats; (**b**) group with 3000 mg/kg ERFE; (**c**) group treated with 2000 mg/kg ERFE; (**d**) group treated with 1000 mg/kg ERFE (magnification 10 × 0.25).

**Table 1 ijms-24-09999-t001:** Total phenolic and flavonoid contents and antioxidant activities of the tested *E. ritro* extracts.

Samples	Total Phenolic Content (mg GAE/g)	Total Flavonoid Content (mg QE)	DPPH (mg TE/g)	ABTS (mg TE/g)	CUPRAC (mg TE/g)	FRAP (mg TE/g)	Chelating (mg EDTAE/g)	Phospho-Molybdenum (mmol TE/g)
ERLE	73.83 ± 0.05	14.52 ± 0.20	120.13 ± 0.38	136.48 ± 1.26	251.67 ± 9.28	153.54 ± 8.27	29.22 ± 0.22	1.52 ± 0.03
ERFE	27.56 ± 0.16	6.37 ± 0.16	44.92 ± 0.07	58.40 ± 0.24	94.89 ± 1.12	54.81 ± 1.76	26.51 ± 0.55	1.24 ± 0.01

Values are reported as mean ± SD of three parallel experiments. GAE: Gallic acid equivalent; QE: quercetin equivalents; TE: Trolox equivalent; EDTAE: EDTA equivalent.

**Table 2 ijms-24-09999-t002:** Serum biochemical parameters in all experimental groups.

Biochemical Parameters	Control	ERFE 300 mg/ kg	Diclofenac 50 mg/kg	ERFE + Diclo	Reference Values [25]
GLU mmol/L	6.39 ± 0.12	7.1 ± 0.42	6.5 ± 0.41	7.8 ± 0.29	6.9–11.3
Urea mmol/L	3.68 ± 0.32	3.8 ± 0.36	4.8 ± 0.22 *	4.6 ± 0.22	3.3–6.8
Creat µmol/L	34.1 ± 12.3	32.3 ± 12.8	35.1 ± 16.2	39.2 ± 12.6	29–81
TP g/L	63.7 ± 3.2	62.2 ± 4.6	62.5 ± 5.3	59.2 ± 3.3	53–63
ALB g/L	28 ± 1.3	28.2 ± 1.2	26.1 ± 2.2	27.4 ± 3.1	26–29
ASAT U/L	86.6 ± 4.1	86.1 ± 5.4	121.3 ± 3.8 *	98.2 ± 4.8	70–155
ALAT U/L	35.2 ± 4.2	37.6 ± 6.1	42.2 ± 4.3	38.4 ± 3.6	35–39
AP U/L	198.3 ± 9.2	162.4 ± 5.3	189.8 ± 6.3	162.4 ± 9.2	62–230
GGT U/L	2.2 ± 0.2	3.2 ± 0.6	5.6 ± 0.3 *	5.8 ± 0.6 *	0.3–6.9
AMYL U/L	599.7 ± 18.2	632.2 ± 26.6	568 ± 12.3	492.3 ± 22.6	485–1942
T-Bil µmol/L	4.4 ± 0.48	4.0 ± 0.28	7.4 ± 0.42 *^+^	4.3 ± 0.38	1.7–6.8
D-Bil µmol/L	3.6 ± 0.86	3.5 ± 0.84	6.4 ± 0.56 *	3.9 ± 0.24	0–6.8
Chol mmol/L	1.1 ± 0.06	2.0 ± 0.09	1.6 ± 0.06	1.7 ± 0.04	1.5–2.1
TRIG mmol/L	0.38 ± 0.001	0.42 ± 0.003	0.66 ± 0.006 *	0.65 ± 0.004 *	1.1–1.2
UA mmol/L	42.6 ± 1.4	38.5 ± 2.3	43.3 ± 3.3	41.9 ± 2.3	30–262

* *p* < 0.05 vs. control group; ^+^ *p* < 0.05 vs. reference range.

**Table 3 ijms-24-09999-t003:** Effect of ERFE pre-treatment on hepatic liver peroxidation and antioxidant profile in rats challenged with diclofenac.

Group	MDA nmol/g Tissue	GSH nmol/g Tissue	GPx nmol/min/mg Protein	GR nmol/min/mg Protein	GST nmol/min/mg Protein
Control	0.385 ± 0.021	6.34 ± 0.38	0.132 ± 0.008	0.220 ± 0.017	0.291 ± 0.014
ERFE	0.369 ± 0.018 *	7.30 ± 0.36 *	0.152 ± 0.006 *	0.258 ± 0.015 *	0.325 ± 0.017 *
Diclofenac	0.486 ± 0.024 *	4.96 ± 0.19 *	0.119 ± 0.005 *	0.173 ± 0.016 *	0.242 ± 0.026 *
ERFE + DF	0.399 ± 0.026 ^+^	6.12 ± 0.25 *^+^	0.138 ± 0.005 ^+^	0.220 ± 0.015 ^+^	0.294 ± 0.023 ^+^

* *p* < 0.05 vs. control group; ^+^ *p* < 0.05 vs. DF-treated group.

**Table 4 ijms-24-09999-t004:** Cytochrome P450, EMND and AH activities measured in rat liver microsomes.

Group	EMND Activity nmol/min/mg	AH Activity nmol/min/mg
Control	0.566 ± 0.018	0.492 ± 0.028
ERFE	0.439 ± 0.034 *	0.422 ± 0.009 *
DF sodium	0.646 ± 0.012 *	0.549 ± 0.019 *
ERFE + DF	0.462 ± 0.009 *^+^	0.448 ± 0.024 *^+^

* *p* < 0.05 vs. control group; ^+^ *p* < 0.05 vs. DF-treated group.

## Data Availability

Not applicable.

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
