# Peer review of "Antioxidant and Hepatoprotective Potential of Echinops ritro L. Extracts on Induced Oxidative Stress In Vitro/In Vivo"

_ijms, 2023, doi:10.3390/ijms24129999_

Round 1

Reviewer 1 Report

The manuscript is aimed to investigate the potential use of Echinops ritro extracts as antioxidant and hepatoprotective agents. The investigation on both of leaves and flowering heads extracts is based on spectrophotometric data and on in vitro and in vivo tests of antioxidant activity and of toxicity. HPLC-MS profiling of the extract metabolites is also described.

The manuscript reports data from a great deal of experimental procedures that appear to be appropriate to the scope of the work. However, there is not a significative innovative apport to what is widely reported in the literature. Echinops ritro extracts have been widely investigated for metabolites content and biological activities. There is a consistent literature already dealing with their antioxidant and potential hepatoprotective activities and the present manuscript does not appear to add any significative innovation.

To my opinion, the manuscript is not appropriate for publication at this stage.

The authors should provide a sound justification of their work, giving clear evidence of innovative results, if any. They should at least fractionate the extracts and identify the most active components.

Besides the references already cited by the authors, also the following papers present scientific data related to the manuscript.

Anvari, Donia; Jamei, Rashid. Evaluation of Antioxidant Capacity and Phenolic Content in Ethanolic Extracts of Leaves and Flowers of Some Asteraceae Species. Recent Patents on Food, Nutrition & Agriculture, Volume 9, Number 1, 2018, pp. 42-49(8). Bentham Science Publishers DOI: https://doi.org/10.2174/2212798409666171023150601

Oksana Sytar, Marek Zivcak, Kiessoun Konate, and Marian Brestic. Phenolic Acid Patterns in Different Plant Species of Families Asteraceae and Lamiaceae: Possible Phylogenetic Relationships and Potential Molecular Markers. Hindawi Journal of Chemistry Volume 2022, Article ID 9632979, https://doi.org/10.1155/2022/9632979

Some English editing is required

Author Response

Response to Reviewer 1 Comments

The manuscript is aimed to investigate the potential use of Echinops ritro extracts as antioxidant and hepatoprotective agents. The investigation on both of leaves and flowering heads extracts is based on spectrophotometric data and on in vitro and in vivo tests of antioxidant activity and of toxicity. HPLC-MS profiling of the extract metabolites is also described. The manuscript reports data from a great deal of experimental procedures that appear to be appropriate to the scope of the work.

However, there is not a significative innovative apport to what is widely reported in the literature. Echinops ritro extracts have been widely investigated for metabolites content and biological activities. There is a consistent literature already dealing with their antioxidant and potential hepatoprotective activities and the present manuscript does not appear to add any significative innovation.

To my opinion, the manuscript is not appropriate for publication at this stage.

Response: Dear reviewer, thanks for the comments. Although Echinops species are drawing increased interest from the phytochemists, much of the research focus has been centered on the thiophenes and terpenes (Bitew and Hymete, 2019). A recent work highlighted a high percentage of mono- and di-caffeoylquinic acids in the aerial parts (leaves and seeds) without differentiation between respective isomers together with scarce number of flavonoids (Zengin et al., 2022). It is worth noting that there is no in-depth study on the E. ritro polyphenolic compounds by hyphenated platform ultra-high-performance liquid chromatography – high resolution mass spectrometry integrated with an assessment of in vitro and in vivo protective effects in induced lipid peroxidation and oxidative stress injury. Despite the accumulating evidence that Echinops extracts reduce liver enzymes and antioxidant markers (Bitew and Hymete, 2019), the effects of any Echinops species on rat liver microsomes and hepatocytes, and histological profil in diclofenac induced liver oxidative stress have not been thoroughly investigated. (See Introduction, rows 95-106).

The presented antioxidant tests including DPPH, ABTS, CUPRAC, FRAP, metal chelating, and phosphomolybdenum methods definitely add inclusiveness to the study. Such a wide-ranging research on the radical scavenging activity of Echinops sp. has not been done and the work upgrades former studies (Erenler et al., 2014; AYDIN et al., 2016; Falah, et al., 2021; Zengin et al., 2022).

Herein, a comprehensive UHPLC-HRMS analysis of leaves and flowering heads was performed yielding the identification/annotation of 108 secondary metabolites. For the first time, 29 phenolic acids and derivatives together with coumarins, 25 acylquinic acids and 41 flavonoid aglycones and glycosides were reported in E. ritro. To the best of our knowledge, phenylethanoid glycosides, hydroxycinnamoyl hexoses (sugar esters), a series of hydroxybenzoic and hydroxicinnamic acid – glycosides, coumaroyl- and feruloylquinic acids, caffeoyl-hydroxydihydrocaffeoylquinic, acids, caffeoyl-feruloylquinic, caffeoyl-dehydrocaffeoylquinic acids, tricaffeoylquinic acid, dicaffeoylquinic acid-hexosides are reported for the first time. Moreover, except for mono-and dicaffeoylquinic acids, the aforementioned acylquinic subclasses are even new compounds for the Echinops genus. In the same manner, flavonoid – rutinosides and acetylhexosides, methoxylated flavonoids and their glycosides represent new secondary metabolites in the species. (See Discussion, rows 472-483).

The abstract was rewritten, presenting the novelty of the work (See Abstract, rows 18-33).

The authors should provide a sound justification of their work, giving clear evidence of innovative results, if any. They should at least fractionate the extracts and identify the most active components.

Besides the references already cited by the authors, also the following papers present scientific data related to the manuscript.

Anvari, Donia; Jamei, Rashid. Evaluation of Antioxidant Capacity and Phenolic Content in Ethanolic Extracts of Leaves and Flowers of Some Asteraceae Species. Recent Patents on Food, Nutrition & Agriculture, Volume 9, Number 1, 2018, pp. 42-49(8). Bentham Science Publishers DOI: https://doi.org/10.2174/2212798409666171023150601

Oksana Sytar, Marek Zivcak, Kiessoun Konate, and Marian Brestic. Phenolic Acid Patterns in Different Plant Species of Families Asteraceae and Lamiaceae: Possible Phylogenetic Relationships and Potential Molecular Markers. Hindawi Journal of Chemistry Volume 2022, Article ID 9632979, https://doi.org/10.1155/2022/9632979

Response: Dear reviewer, thanks for the recommendations. New paragraphs were embedded into the text, concerning novelty of the work (See Abstract rows 18-33, Introduction, rows 95-106, and Discussion, rows 472-483).

A previous study on E. ritro has reported important results applying multivariate data analysis to determine correlation between chemical composition and antioxidant activity in different tests (Zengin et al., 2022). Thus, radical scavenging and reducing capacity positively correlated with the major compounds, neochlorogenic/chlorogenic acid, quercetin/hesperidin-hexoside and dicaffeoylquinic acids, while metal chelating activity was associated with apigenin O-hexuronide and total antioxidant capacity – with naringenin-coumaroylhexoside. From the other hand, the crude E. orientalis Trautv. seeds and leaves extracts showed more pronounced radical scavenging effects (> 60% at 40 µg/mL) compared to the isolated flavonoids apigenin 7-O-glucoside and apigenin 7-O-(6"-trans p-coumaroyl)-β-D-glucopyranosode (Erenler et al., 2014).

It is worth noting that the antioxidant activity is usually dose-dependent as has been seen in the E. albicaulis aerial parts study and at higher concentration overproduction of ROS was observed (Kiyekbayeva et al., 2017). (See Discussion, row 433-471).

The papers presented scientific data related to the manuscript (Jamei and Anvari, 2018] and Sytar et al., 2022) were included (See Introduction, row 72).

They should at least fractionate the extracts and identify the most active components.

Response: Dear reviewer, thanks for the recommendations. The aim of the study was to evaluate the potential of E. ritro leaves and flowering heads extracts as antioxidant and hepatoprotective agents in induced lipid peroxidation and oxidative stress under in vitro and in vivo conditions. In addition, the profiling of the tested extracts was performed using UHPLC-HRMS. In this study we are not focus on the bioactive guided fractionation and isolation of the main compounds. The predominant secondary metabolites contribute to the antioxidant and hepatopretective activity were presented (See Results, rows 269-276; rows 293-300; rows 335-338).

The following paragraphs were embedded into the text: “The extracted ion chromatograms of phenolic acids and derivatives showed that the E. ritro leaves profile was dominated by protocatechuic acid-O-hexoside (1) (10.09%), gentisic acid (27) (2.26%) and protocatechuic acid-(salicyl)-hexoside (32) (3.05%) together with quinic acid (17) (16.24 %). Among acylquinic acid, the predominant compounds in both, leaves and flower heads were neochlorogenic (34) (6.59% in ERLE, 0.67% in ERFE), chlorogenic (37) (18.28% in ERLE, 9.35% in ERLE), 4-caffeoylquinic (38) (17.52% in ERLE, 9.29% in ERFE) and 3, 5-dicaffeoylquinic acid (53) (9.90% in ERLE, 18.44% in ERFE) (Figure S1).” (See Results, rows 269-276).

“Flavonoid glycosides profile of the leaves extract was dominated by isoquer-citrin (64) (11.88%), hyperoside (67) (11.38%) together with luteolin-O-hexuronide (68) (11.28) rutin (63) (6.39%), kaempferol 3-O-rutinoside (72) (4.50%), astragalin (75) (4.50%) and apigenin 7-O-hexuronide (77) (3.92%). In contrast, 77 (5.57%) is prevailing flavonoid glycoside in the flower heads extract accompanied by apig-enin O-rutinoside (74) (2.63%) and hyperoside (67) (1.12%) (Figure S2). Previously, apigenin was determined in E. echinatus, E. spinosus, and E. orientalis [11], while luteolin 7-O-glucoside was found in E. spinosus [32].” (See Results, rows 293-300).

Overall, eupatilin (105) (4.60%) together with cirsiliol (92) (3.85%) and ja-ceosidin (99) (3.71%) appeared to be characteristic for leaves extracts; in contrast, flower heads were the richest in apigenin (90) (7.47%), chrysoeriol (93) (3.55%), and luteolin (87) (3.33%), (Figure S3). (See results, rows 335-338).

Some English editing is required

Response: Dear reviewer, thanks for the comment. The English of the whole manuscript was edited.

Reviewer 2 Report

1.      The method for UHPLC-HRMS mass spectrometry analyses need to be briefly described (Page18, Line572) .

2.      The authors are encouraged to display the content of secondary metabolites identified in both ERLE and ERFE extracts (Page 8, Line230) .

3.      How does it come to the result “Furthermore, the ERFE beneficial effect could be also associated to its metabolite-mediated activities and inhibitory role on the diclofenac bioactivation and toxic metabolites formation. A variety of hydroxybenzoic, hydroxycinnamic, acylquinic acids, phenylethanoid glycsides, and flavonoids in the flowering heads and leaves extracts could be related to its antioxidant and hepatopretective activity”.(Page19,Line 586-589)?

Hunderds of secondary metabolites were identified in both ERFE and ERLE extracts, however, it is hard to come to the conclusion that hydroxybenzoic, hydroxycinnamic, acylquinic acids, phenylethanoid glycsides, and flavonoids in the flowering heads and leaves extracts could be related to its antioxidant and hepatopretective activity.

4.      Which are the predominant secondary metabolites contribute to the antioxidant and hepatopretective activity? there is no enough data to support it. Please revise.

5.      Kindly revise reference format according to the author guideline.

6.      It is suggested to cite references within 5 years of research to maintain the reliability of results obtained.

Minor editing of English language required

Author Response

Response to Reviewer 2 Comments

1.The method for UHPLC-HRMS mass spectrometry analyses need to be briefly described (Page18, Line572) .

Response: Dear reviewer, thanks for the comment. The used UHPLC-HRMS method was briefly described (See Materials and methods, rows 645-653).

  1. The authors are encouraged to display the content of secondary metabolites identified in both ERLE and ERFE extracts (Page 8, Line230) .

Response: Dear reviewer, thanks for the comment. The following paragraphs were embedded into the text: “The extracted ion chromatograms of phenolic acids and derivatives showed that the E. ritro leaves profile was dominated by protocatechuic acid-O-hexoside (1) (10.09%), gentisic acid (27) (2.26%) and protocatechuic acid-(salicyl)-hexoside (32) (3.05%) together with quinic acid (17) (16.24 %). Among acylquinic acid, the predominant compounds in both, leaves and flower heads were neochlorogenic (34) (6.59% in ERLE, 0.67% in ERFE), chlorogenic (37) (18.28% in ERLE, 9.35% in ERLE), 4-caffeoylquinic (38) (17.52% in ERLE, 9.29% in ERFE) and 3, 5-dicaffeoylquinic acid (53) (9.90% in ERLE, 18.44% in ERFE) (Figure S1).” (See Results, rows 269-276).

“Flavonoid glycosides profile of the leaves extract was dominated by isoquer-citrin (64) (11.88%), hyperoside (67) (11.38%) together with luteolin-O-hexuronide (68) (11.28) rutin (63) (6.39%), kaempferol 3-O-rutinoside (72) (4.50%), astragalin (75) (4.50%) and apigenin 7-O-hexuronide (77) (3.92%). In contrast, 77 (5.57%) is prevailing flavonoid glycoside in the flower heads extract accompanied by apig-enin O-rutinoside (74) (2.63%) and hyperoside (67) (1.12%) (Figure S2). Previously, apigenin was determined in E. echinatus, E. spinosus, and E. orientalis [11], while luteolin 7-O-glucoside was found in E. spinosus [32].” (See Results, rows 293-300).

Overall, eupatilin (105) (4.60%) together with cirsiliol (92) (3.85%) and ja-ceosidin (99) (3.71%) appeared to be characteristic for leaves extracts; in contrast, flower heads were the richest in apigenin (90) (7.47%), chrysoeriol (93) (3.55%), and luteolin (87) (3.33%), (Figure S3). (See results, rows 335-338).

MZmine 2 software was applied to the UHPLC–HRMS raw files of the studied E. ritro extracts for the semi-quantitative analysis. Results are expressed as % peak area of the compound to the total peaks areas of the corresponding group secondary metabolites.

  1. How does it come to the result “Furthermore, the ERFE beneficial effect could be also associated to its metabolite-mediated activities and inhibitory role on the diclofenac bioactivation and toxic metabolites formation? A variety of hydroxybenzoic, hydroxycinnamic, acylquinic acids, phenylethanoid glycsides, and flavonoids in the flowering heads and leaves extracts could be related to its antioxidant and hepatopretective activity”. (Page19, Line 586-589)?

Hundreds of secondary metabolites were identified in both ERFE and ERLE extracts, however, it is hard to come to the conclusion that hydroxybenzoic, hydroxycinnamic, acylquinic acids, phenylethanoid glycsides, and flavonoids in the flowering heads and leaves extracts could be related to its antioxidant and hepatopretective activity.

Response: Dear reviewer, thanks for the question. All of these secondary metabolites could be related to antioxidant and hepatopretective activity because World scientific literature abounds with data on these beneficial effects of hydroxybenzoic, hydroxycinnamic, acylquinic acids, phenylethanoid glycosides, and flavonoids. At this stage of the experiment, we cannot say exactly which bioactive compound is exhibiting these effects. This would be the subject of future additional work studying the pharmacological and therapeutic potential of each identified compound in the extract. It is likely that the combination of all these compounds in the extract is related to their hepatoprotective and antioxidant effects.There is evidence both for their independent hepatoprotective and antioxidant potential and for their synergistic or potentiating action in terms of liver protection through their antioxidant capacity.

Recently Vukmirović et al., (Vukmirović, S.; Ilić, V.; Tadić, V.; ÄŒapo, I.; Pavlović, N.; Tomas, A.; Paut Kusturica, M.; Tomić, N.; Maksimović, S.; Stilinović, N. Comprehensive Analysis of Antioxidant and Hepatoprotective Properties of Morus nigra L. Antioxidants 2023, 12, 382. https://doi.org/10.3390/antiox12020382)  investigated the chemical composition, antioxidant and hepatoprotective properties of Morus nigra L. in vivo in STZ- induced oxidative stress in NMRI Haan mice. Authors discovered that out of the phenolic acids, predominant were derivatives of hydroxycinnamic acid—chlorogenic acid, derivatives of hydroxybenzoic acid—gallic acid and derivatives of dihydroxybenzoic acid—vanillic acid. In terms of the flavonoid content, examined black mulberry extracts were abundant in flavones and flavonols, among which, the most abundant in analyzed extracts were quercetin, kaempferol, isoquercetin, hyperoside and rutin and predominately being isoquercetin and hyperoside. Due to the higher content of phenolic compounds, leaf and bark ethanol extracts were found to possess a stronger antioxidant and hepatoprotective effect. Additionally, the significantly weaker activity of the P450 enzymes was detected in the black mulberry leaf and bark ethanol extracts, which is an indirect indicator of a better recovery of liver tissue. These findings support our suggestion that inhibiting CYP 450 isoenzymes activity (AH and EMND) could decrease diclofenac-induced toxicity, which we have established in our experiment.

Chakrabarty  al. (Chakrabarty, N.; Chung, H.-J.; Alam, R.; Emon, N.U.; Alam, S.; Kabir, M.F.; Islam, M.M.; Hong, S.-T.; Sarkar, T.; Sarker, M.M.R.; Rahman, M.M. Chemico-Pharmacological Screening of the Methanol Extract of Gynura nepalensis D.C. Deciphered Promising Antioxidant and Hepatoprotective Potentials: Evidenced from in vitro, in vivo, and Computer-Aided Studies. Molecules 2022, 27, 3474. https://doi.org/10.3390/molecules27113474) performed a chemico-pharmacological screening of the methanol extract of Gynura nepalensis. This in vitro, in vivo, and in silico study discovered the beneficial effects of 4,5-dicaffeoylquinic acid and chlorogenic acid. Hepatoprotective and antioxidant molecular docking studies found that these chemical components revealed the highest binding affinity among the selected molecules.  It is presumed that the hepatoprotective properties of the plant extract have occurred due to the presence of these bioactive chemical compounds as well as their antioxidant properties.

In our previous investigation (Simeonova R, Vitcheva V, Zheleva-Dimitrova D, Balabanova V, Savov I, Yagi S, Dimitrova B, Voynikov Y, Gevrenova R. Trans-3,5-dicaffeoylquinic acid from Geigeria alata Benth. & Hook.f. ex Oliv. & Hiern with beneficial effects on experimental diabetes in animal model of essential hypertension. Food Chem Toxicol. 2019;132:110678. https://doi.org/10.1016/j.fct.2019.110678) of Geigeria alata Benth. & Hook.f. ex Oliv. & Hiern we discovered that trans-3,5-dicaffeoylquinic acid possess a potent antioxidant, antidiabetic and hepatoprotective effects.

In the study of Tian et al., (Tian, Y.; Xia, T.; Qiang, X.; Zhao, Y.; Li, S.; Wang, Y.; Zheng, Y.; Yu, J.; Wang, J.; Wang, M. Nutrition, Bioactive Components, and Hepatoprotective Activity of Fruit Vinegar Produced from Ningxia Wolfberry. Molecules 2022, 27, 4422. https://doi.org/10.3390/molecules27144422) bioactive compounds, and hepaprotective activity in wolfberry vinegar (WFV) were explored. p-Hydroxybenzoic acid and m-hydroxycinnamic acid were the main polyphenols in WFV. WFV treatment effectively alleviated liver injury in CCl4-treated mice by improving histopathological changes and reducing liver biochemical indexes, and decreased oxidative damage by inhibiting oxidative status and increasing antioxidant potential.

De Camargo et al., (de Camargo, A.C.; Concepción Alvarez, A.; Arias-Santé, M.F.; Oyarzún, J.E.; Andia, M.E.; Uribe, S.; Núñez Pizarro, P.; Bustos, S.M.; Schwember, A.R.; Shahidi, F.; Bridi, R. Soluble Free, Esterified and Insoluble-Bound Phenolic Antioxidants from Chickpeas Prevent Cytotoxicity in Human Hepatoma HuH-7 Cells Induced by Peroxyl Radicals. Antioxidants 2022, 11, 1139. https://doi.org/10.3390/antiox11061139) discovered that m-hydroxybenzoic acid, taxifolin, and biochanin A were the main phenolics found in chickpeas. m-hydroxybenzoic acid was present mainly in the insoluble-bound form. The insoluble-bound fraction made a significant contribution to the reducing power and antiradical activity towards peroxyl radical. Furthermore, the extract from chickpeas decreased the oxidative damage of human HuH-7 cells induced by peroxyl radicals, thus indicating its hepatoprotective potential.

The study of Fan et al.,(Fan, Y.-C.; Yue, S.-J.; Guo, Z.-L.; Xin, L.-T.; Wang, C.-Y.; Zhao, D.-L.; Guan, H.-S.; Wang, C.-Y. Phytochemical Composition, Hepatoprotective, and Antioxidant Activities of Phyllodium pulchellum (L.) Desv. Molecules 2018, 23, 1361. https://doi.org/10.3390/molecules23061361) revealed that the flavonoids isolated from Phyllodium pulchellum (L.) Desv, showed hepatoprotective and antioxidant activities, indicating that, besides alkaloids, the flavonoids should be the potential pharmacodynamic ingredients that are responsible for the hepatoprotective and antioxidant activities of this species.

  1. Which are the predominant secondary metabolites contribute to the antioxidant and hepatopretective activity? There is no enough data to support it. Please revise.

Response: Dear reviewer, thanks for the question. The predominant secondary metabolites contribute to the antioxidant and hepatopretective activity were presented (See Results, rows 269-276; rows 293-300; rows 335-338; Discussion, rows 433-463).

  1. Kindly revise reference format according to the author guideline.

Response: Dear reviewer, thanks for the comment. The references were changed according to the IJMS requirements.

  1. It is suggested to cite references within 5 years of research to maintain the reliability of results obtained.

Response: Dear reviewer, thanks for the comment. Most of the older citations in the manuscript are related to the pharmacological and toxicological methods used, which are established in scientific circles and applied by many research groups.

We have added new references from the last five years:

Vukmirović, S.; Ilić, V.; Tadić, V.; Čapo, I.; Pavlović, N.; Tomas, A.; Paut Kusturica, M.; Tomić, N.; Maksimović, S.; Stilinović, N. Comprehensive Analysis of Antioxidant and Hepatoprotective Properties of Morus nigra L. Antioxidants 2023, 12, 382. https://doi.org/10.3390/antiox12020382;

Chakrabarty, N.; Chung, H.-J.; Alam, R.; Emon, N.U.; Alam, S.; Kabir, M.F.; Islam, M.M.; Hong, S.-T.; Sarkar, T.; Sarker, M.M.R.; Rahman, M.M. Chemico-Pharmacological Screening of the Methanol Extract of Gynura nepalensis D.C. Deciphered Promising Antioxidant and Hepatoprotective Potentials: Evidenced from in vitro, in vivo, and Computer-Aided Studies. Molecules 2022, 27, 3474. https://doi.org/10.3390/molecules27113474;

Tian, Y.; Xia, T.; Qiang, X.; Zhao, Y.; Li, S.; Wang, Y.; Zheng, Y.; Yu, J.; Wang, J.; Wang, M. Nutrition, Bioactive Components, and Hepatoprotective Activity of Fruit Vinegar Produced from Ningxia Wolfberry. Molecules 2022, 27, 4422. https://doi.org/10.3390/molecules27144422;

de Camargo, A.C.; Concepción Alvarez, A.; Arias-Santé, M.F.; Oyarzún, J.E.; Andia, M.E.; Uribe, S.; Núñez Pizarro, P.; Bustos, S.M.; Schwember, A.R.; Shahidi, F.; Bridi, R. Soluble Free, Esterified and Insoluble-Bound Phenolic Antioxidants from Chickpeas Prevent Cytotoxicity in Human Hepatoma HuH-7 Cells Induced by Peroxyl Radicals. Antioxidants 2022, 11, 1139. https://doi.org/10.3390/antiox11061139;

Sweilam, S.H.; Abdel Bar, F.M.; Foudah, A.I.; Alqarni, M.H.; Elattal, N.A.; El-Gindi, O.D.; El-Sherei, M.M.; Abdel-Sattar, E. Phytochemical, Antimicrobial, Antioxidant, and In Vitro Cytotoxicity Evaluation of Echinops erinaceus Kit Tan. Separations 2022, 9, 447. https://doi.org/10.3390/separations9120447;

Simeonova R, Vitcheva V, Zheleva-Dimitrova D, Balabanova V, Savov I, Yagi S, Dimitrova B, Voynikov Y, Gevrenova R. Trans-3,5-dicaffeoylquinic acid from Geigeria alata Benth. & Hook.f. ex Oliv. & Hiern with beneficial effects on experimental diabetes in animal model of essential hypertension. Food Chem Toxicol. 2019;132:110678. https://doi.org/10.1016/j.fct.2019.110678;

Fan, Y.-C.; Yue, S.-J.; Guo, Z.-L.; Xin, L.-T.; Wang, C.-Y.; Zhao, D.-L.; Guan, H.-S.; Wang, C.-Y. Phytochemical Composition, Hepatoprotective, and Antioxidant Activities of Phyllodium pulchellum (L.) Desv. Molecules 2018, 23, 1361. https://doi.org/10.3390/molecules23061361;

Besides the references already cited by the authors, also the following papers present scientific data related to the manuscript.

Anvari, Donia; Jamei, Rashid. Evaluation of Antioxidant Capacity and Phenolic Content in Ethanolic Extracts of Leaves and Flowers of Some Asteraceae Species. Recent Patents on Food, Nutrition & Agriculture, Volume 9, Number 1, 2018, pp. 42-49(8). Bentham Science Publishers DOI: https://doi.org/10.2174/2212798409666171023150601

Oksana Sytar, Marek Zivcak, Kiessoun Konate, and Marian Brestic. Phenolic Acid Patterns in Different Plant Species of Families Asteraceae and Lamiaceae: Possible Phylogenetic Relationships and Potential Molecular Markers. Hindawi Journal of Chemistry Volume 2022, Article ID 9632979, https://doi.org/10.1155/2022/9632979

Minor editing of English language required

Response: Dear reviewer, thanks for the comment. The English language of the whole manuscript was edited.

Round 2

Reviewer 1 Report

The authors satisfied the main observations of the first review.